# Survival of vascularized osseous flaps in mandibular reconstruction: A network meta-analysis

**Mubarak Ahmed Mashrah** [1,2‡] *, **Taghrid Aldhohrah** [3‡], **Ahmed Abdelrehem** [4], **Karim Ahmed Sakran** [5], **Hyat Ahmad** [6], **Hamada Mahran** [7], **Faisal Abu-Iohom** [2], **Hanfu Su** [1], **Ying Fang** [1] *, **Liping Wang** [1] *

**1** Key Laboratory of Oral Medicine, Guangzhou Institute of Oral Disease, Stomatology Hospital of Guangzhou Medical University, Guangdong, Guangzhou, China, **2** Jibla University for Medical Sciences, Jibla Hospital, Ibb City, Yemen, **3** Guanghua Stomatology Hospital, Sun Yat-Sen University, Guangdong, Guangzhou, China, **4** Department of Craniomaxillofacial and Plastic Surgery, Faculty of Dentistry, Alexandria University, Alexandria, Egypt, **5** State Key Laboratory of Oral Diseases and National Clinical Research Center for Oral Diseases and Department of Oral and Maxillofacial Surgery, West China Hospital of Stomatology, Sichuan University, Chengdu, Sichuan, China, **6** Department of Oral Pathology, Dalian Medical University, China, **7** Maxillofacial Surgery Unit, General Surgery Department, Faculty of Medicine, Assiut University, Assiut, Egypt

‡ These authors share first authority on this work.
* mubarak198226@gmail.com (MAM); 10747936@qq.com (YF); wangliplj@126.com (LW)

**Data Availability Statement:** The authors confirm that all data underlying the findings are fully available without restriction. All relevant data are

## Abstract

### Objective

An evidence regarding which bony flap for reconstruction of mandibular defects following tumour resection is associated with the highest survival rate is still lacking. This network meta-analysis (NMA) aimed to guide surgeons selecting which vascularized osseous flap is associated with the highest survival rate for mandibular reconstruction.

### Methods

From inception to March 2021, PubMed, Embase, Scopus, and Cochrane library were searched to identify the eligible studies. The outcome variable was the flap survival rate. The Bayesian NMA accompanied by a random effect model and 95% credible intervals (CrI) was calculated.

### Results

Twenty-two studies with a total of 1513 patients, comparing four osseous flaps namely fibula free flap (FFF), deep circumferential iliac artery flap (DCIA), scapula flap, and osteocutaneous radial forearm flap (ORFF) were included. The respective survival rates of FFF, DCIA, Scapula, and ORFF were 94.50%, 93.12%, 97%, and 95.95%. The NMA failed to show a statistically significant difference between all comparators (FFF versus DCIA (Odd ratio, 1.8; CrI, 0.58,5.0); FFF versus ORFF (Odd ratio, 0.57; CrI, 0.077; 2.9); FFF versus scapula flap (Odd ratio, 0.25; CrI, 0.026; 1.5); DCIA versus ORFF (Odd ratio, 0.32; CrI,

within the paper and its Supporting Information
files.

**Funding:** The authors received no specific funding
for this work.

**Competing interests:** The authors reported no
conflicts of interest related to this study.

0.037; 2.1); DCIA versus scapula flap (Odd ratio, 0.14; CrI, 0.015; 1.1) and ORFF versus
scapula flap (Odd ratio, 2.3; CrI, 0.16; 34)).

## Conclusion

Within the limitations of the current NMA, FFF, DCIA, Scapula, and ORFF showed a compa-
rable survival rate for mandibular reconstruction. Although the scapula flap reported the
highest survival rate compared to other osseous flaps for mandibular reconstruction; how-
ever, the decision making when choosing an osseous flap should be based on many factors
rather than simply flap survival rate.

## 1. Introduction

Mandibular defects resulting from severe trauma or post-oncologic resection often led to sig-
nificant functional and aesthetic limitations. Therefore, the fundamental principles of mandib-
ular reconstruction should be directed to restore the form, function, and aesthetics. Critical to
this is the restitution of the 3-dimensional anatomical relationship by restoring the mandible's
continuity, contour, vertical height, and alveolar ridge that is conducive to dental implant
placement and prosthetic rehabilitation [1–3]. The evolution of microsurgery allows the appli-
cation of vascularized bone flaps in a single-staged primary mandibular reconstruction [4].

Composite free tissue transfer is nowadays considered the standard treatment tool in recon-
structive surgery following various mandibular defects [5–9]. The most frequently used osse-
ous free flaps are harvested from the fibula [10, 11], iliac crest [12, 13], scapula [14, 15] and
radius [16] in order of frequency of use. Each flap possesses its unique properties, including
the length and size of the vascular pedicle, quantity and quality of the osseous component,
associated soft tissue versatility, donor morbidity, possibility of osteotomies and suitability for
reshaping to mimic the parabolic shape of the mandible, and the feasibility of dental implant
placement [16].

The choice of flaps is determined by several factors including, the timing of reconstruction,
the recipient site conditions, the location of the defect, and the amount of bone and soft tissue
required. In general, there is no agreement that an individual vascularized osseous flap could
be the choice to restore all classes of mandibular defects [1, 16]. A suitable flap, therefore,
should be selected according to an algorithm defining the specific type of bone and soft tissue
defects. For proper flap selection, the bony defect condition should be considered first, fol-
lowed by the soft tissue. When the bony defect is "lateral" and the soft tissue is not defective,
the ilium is the best choice. When the bony defect is "lateral" and a small "skin or mucosal"
soft-tissue defect is present, the fibula represents the optimal choice. When the bony defect is
"lateral" and an extensive "skin or mucosal" or "through-and-through" soft-tissue defect exists,
the scapula should be selected. When the bony defect is "anterior," the fibula is then the pre-
ferred choice. However, when an "anterior" bone defect also displays an "extensive" or
"through-and-through" soft-tissue defect, the fibula should be used with other soft-tissue flaps
[17, 18]. However, the clinical implementation of the currently available treatment algorithm
is still incomplete. Reviewing the literature, only two conventional meta-analyses reviewing
the survival of such flaps were found [19, 20], however, both did not answer the question
regarding which osseous flap has a higher survival rate for mandibular reconstruction.

Due to lack of the evidence regarding which bony flap is associated with the highest survival
rate, therefore, a comprehensive comparative study of different osseous flaps for mandibular

reconstruction is mandated. To date, all published studies provide "head-to-head" or direct comparison for only 2 or three osseous flaps; however, no study compared the above-mentioned osseous flaps has been published yet. Network meta-analyses (NMA) have gained a wide popularity in comparing two treatments that have not been compared directly in a head-to-head clinical trial, thus potentially facilitating timely recommendations and reducing research waste [21].

The current NMA aimed to summarize the available evidence and to answer the question; which vascularized osseous flap is associated with the highest survival rate for mandibular reconstruction.

## 2. Materials and methods

The current meta-analysis was conducted according to the Preferred Reporting Items for the PRISMA Extension Statement for Reporting of Systematic Reviews Incorporating Network Meta-Analyses of Health Care Interventions [22]. (S1 Table) The protocol of the current NMA was registered in the PROSPERO platform (CRD42020207777).

### 2.1. Focused question

Which vascularized osseous flap is associated with the highest survival for mandibular reconstruction? The question for the current meta-analysis was adopted to follow the PICO criteria:

**P:** Patients with mandibular tumor and underwent mandibulectomy.

**I:** Participants who received mandibular reconstruction using the vascularized osseous flaps.

**C:** Different vascularized bony flaps (fibula free flap (FFF), deep circumflex iliac artery (DCIA) flap, scapula flap and osteocutaneous radial free flap (ORFF)).

**O:** Flap survival rate.

### 2.2. Search strategy

From inception to March 2021, an electronic search on Medline/PubMed, EMBASE, Cochrane Central, and Scopus was performed by two reviewers independently (S2 Table). The following keywords were used for the electronic search in PubMed: ((((((mandibl*)) OR (oral cancer[MeSH Terms])) OR (bone tissue neoplasms[MeSH Terms])) OR (mandibulectomy)) AND ((((free flap[MeSH Terms]) OR (free flaps, microsurgical[MeSH Terms])) OR (osseous flap)) OR (bony flap))) AND (((((((fibula[MeSH Terms]) OR (scapula[MeSH Terms])) OR (radius[MeSH Terms])) OR (DCIA FLAP)) OR (osteocutaneous fibula flap)) OR (deep circumflex iliac artery)) OR (radialforearm flap))) AND (((survival rate[MeSH Terms]) OR (survival rates[MeSH Terms])) OR (flap failure)).

A manual search in different dental journals (Laryngoscope, Head and neck, JAMA otolaryngology head and neck surgery, International Journal of Oral and Maxillofacial Surgery, European Journal of craniomaxillofacial surgery, British Journal of oral and maxillofacial surgery, Journal of Plastic and reconstructive surgery, and Aesthetic, Plastic and reconstructive surgery, Microsurgery, and annals of plastic surgery) was also carried out. In addition, the references of the related articles were carefully checked for studies that met the inclusion criteria.

### 2.3. Inclusion and exclusion criteria

The inclusion criteria were as follows: (1) Randomized and non- randomized controlled clinical trials with at least 5 participants in each group; (2) Studies that compared two or more

vascularized osseous flaps (FFF, Scapula flap, DCIA, and ORFF) for mandibular reconstruction; (3) Studies published in English Language and reporting flap survival rate.

The exclusion criteria were as follows: (1) Studies reporting the use of osseous flaps for reconstruction of sites other than the mandible; (2) Animal studies, case series, and review articles were excluded from this study; (3) Studies including less than 5 patients in each group or did not report flap survival rate.

## 2.4. Data extraction process

Two researchers independently assessed the relevant studies (titles, abstracts, and full-text) and any controversy was resolved by discussion to reach a common consensus. The following data (authors, publication year, country of origin, study design, number of patients, age of participants, type of flaps, flap failure, preoperative radiotherapy and follow-up period, and other outcomes) were collected from each study when available (Table 1). Two researchers independently assessed the included studies and collected the data regarding the outcomes of interest. Disagreements between the two researchers were solved by a third reviewer.

## 2.5. Risk of bias assessment

The risk of bias was independently assessed by two authors. Quality assessment of the risk of bias for RCTs was carried out using Cochrane collaboration's tool [23], whereas the Newcastle-*Ottawa* Scale (NOS) was used for assessment of the non-RCTs. The RCTs were evaluated using the following six items: random sequence generation, allocation concealment, blinding of outcome assessment, incomplete outcome data, selected reporting, and other bias. If a particular study met all the above criteria, the study was then rated as low risk of bias. If one or more of the above domains were unclear, the study was considered as unclear risk of bias. If one or more of these criteria were not met, the study was classified as having a high risk of bias. The NOS scale was used for assessment of the quality of the non-RCTs, 4 points for selection, 2 points for comparability, and 3 points for the outcome. Studies gained 6–9 points were classified as high quality. In case there was a disagreement, a third reviewer was consulted (S3 Table).

## 2.6. Data synthesis

Traditional pairwise meta-analyses for direct comparison were firstly performed using A Comprehensive Meta-Analysis Software (Biostat Inc, Englewood, NJ). Odds ratio along with 95% CI was calculated. The heterogeneity across the included studies was evaluated using the Cochrane Q test ($\chi2$ test) and I-squared index (I2). If I 2 was between 0% to 25%, no heterogeneity; if I 2 = 25% to 50%, moderate heterogeneity; if I 2 = 50% to 75%, high heterogeneity whereas I 2 = 75% to 100%, extreme heterogeneity [24]. When I2>50%, the random effect model was used [25], while in case of I2<50%, a fixed effect model was used. The p-value of <0.05 was considered statistically significant.

The network map was drawn and representing a graphical depiction of all direct comparisons whereby the thickness of lines between nodes represented the number of direct comparisons in the included studies. The NMA outputs for binary data were Odd ratio accompanied by 95% confidence intervals (CrI). The rank probability was used to determine which intervention is the best, the second-best, etc. under the posterior distribution derived from the relative effect. A node-splitting analysis was applied to determine the accuracy of indirect comparative estimates from the available direct comparisons. Comparison-adjusted funnel plots were inspected visually for publication bias. The whole NMA was performed using a Bayesian

**Table 1. General characteristics of the included studies.**

| Author Year Country | Study design | Age (Year) | Total number of patients | Comparisons | Classification of Mandibular defect | No. of flap loss | Follow-up | Other outcomes | Bias assessment |
|---|---|---|---|---|---|---|---|---|---|
| Ritschl et al. 2020 Germany | RS | NR | 113 | Fibula = 89 DCIA = 24 | NR | 4 | 32 months (12–63 months) | Fistula formation, Dehiscence, bone exposure; | Moderate |
| Haughey et al. 1994 USA | RS | Median = 62.5; range 16–79 | 21 | Fibula = 9 DCIA = 12 | NR | 1 | | Implant survival; | Low |
| Heller et al. 1995 USA | RS | Median = 67; ranged 13 to 82 | 73 | Fibula = 5 DCIA = 16 Scapula = 2 Non-vascularized iliac = 2 | NR | 2 | 14 | Infection; hospital stay; Quality of life; | Moderate |
| Schultz et al. 2015 USA | RS | 52.2 years (range, 15 to 64 years) | 24 | Fibula = 19 DCIA = 5 | Type 1 = 5 Type 2 = 12 Type 3 = 6 Type 4 = 1 (#) | 1 | 0.5–84 months | Donor site complication, infection fistula formation | Low |
| Shpitzer et al. 1985 Canada | RS | 19 to 85 years | 117 | Fibula = 59 DCIA = 58 | H11 = L = 29 C = 6 HC = 5 LC = 39 LCL = 27 HCL = 3 | 8 | mean 18 months | Systemic, donor site, recipient site complications | Low |
| Takushima et al. 2001 Japan | RS | 55 (13–85) Years | 176 | Fibula = 34 DCIA = 36 Scapula = 51 | H = 2 L = 54 HC = 3 LC = 70 LCL = 49 | 13 | | Infection, fistula; and Postoperative recipient site functional outcomes | Low |
| Chang et al. 2001 USA | RS | mean 55.4 (41 to 74) years | 29 | Fibula = 17 DCIA = 5 (Scapula = 2; ORFF = 1; Rectus abdominis myocutaneous = 4) | NR | 4 | Mean 2 years 9 months (range, 5 months to 7 years 8 months) | Skin paddle loss; other complications | Moderate |
| Yilmaz et al. 2008 Turky | RS | 38 years ranging from 12 to 71 | 37 | DCIA = 24 Fibula = 13 | NR | 1 | 5.72 months | Functional and aesthetic outcomes | Moderate |
| Van Germert et al. 2011 Netherlands | RS | Mean = 61.4 ranged (23.6–84.9) | 83 | fibula flap n = 46 iliac crest flap n = 22 | NR | 6 | | Hematoma, wound dehesence, plate exposure, fistula | Low |
| Boyd et al. 1990 Canada | RS | range: 18 to 85 years | 73 | DCIA = 60 ORFF = 13 | NR | 3 | Mean 9 months (1 month to 4 years) | Early complication (infection, fistula, hematoma) malunion, hardwere exposure, donor site. Hospital stay | Low |
| Virgin et al. 2010 USA | RS | Mean ORFF = 63.7 Fibula = 59 years | 168 | ORFF = 117 Fibula = 51 | NR | 6 | 25.5 months | Functional Outcomes of mandible; Malunion and Donor site complications | Low |
| Dean et al. 2011 USA | RS | Mean 66.5 (range, 31–96) years | 124 | ORFF = 73 Fibula = 51 | | 5 | Mean 17 months | Early complication (infection, fistula, hematoma) malunion, hardwere exposure, donor site. Hospital stay | Low |

*(Continued)*

**Table 1.** (Continued)

| Author Year Country | Study design | Age (Year) | Total number of patients | Comparisons | Classification of Mandibular defect | No. of flap loss | Follow-up | Other outcomes | Bias assessment |
|---|---|---|---|---|---|---|---|---|---|
| Fujiki et al. 2013 Japan | RS | 24–80 years | 46 | Fibula = 38 Scapula = 18 | L = 29 LC = 18 LCL = 8 | 2 | NR | Operative time, Systemic complications; Recipient site complications; Partial flap loss; Wound infection; Fistula formation; Wound dehiscence; Seroma; Haematoma and donor site complications | Low |
| Dowthwaite et al. 2013 Canada | RS | Mean 62 years | 110 | Fibula = 58 Scapula = 55 | NR | 1 | NR | Operative time; Hardware exposure; Nonunion/ malunion; Donor-site complication; Wound breakdown; | Low |
| Deleyiannis et al. 2006 USA | RS | NR | 48 | ORFF = 31 Fibula = 8 | Type I = 60 Type II = 11 Type III = 5 (**) | 3 | NR | Medical complications | Moderate |
| Hanken et al. 2014 Germany | RS | Mean 53.3 (31 to 76) year | 30 | Fibula = 25 DCIA = 5 | LCL = 5 LC = 2 C = 4 L = 14 H = 1 (*) | 1 | Mean = 383 days | Duration of operation, ICU and hospital stay | Moderate |
| Chen et al. 2014 Taiwan | | Average Fibula = 50.8 DCIA = 52.1 years | 153 | Fibula = 45 DCIA = 108 | NR | 0 | NR | Postoperative infection rate, nonunion/malunion rate, mean hospital stay, and antibiotics use | Moderate |
| Chen et al. 1994 Taiwan | RS | NR | 55 | Fibula = 20 DCIA = 32 Scapula = 3 | NR | 1 | NR | Donor site morbidity and scar Bony union Osteointegration | Moderate |
| Yu et al. 2019 China | RS | Mean 38.4(11.6) | 30 | Fibula = 10 DCIA = 20 | NR | 0 | NR | Hight and valium comparison of both flaps | Moderate |
| Politi, and Toro. 2012 Italy | RCT | Mean = 56 Range (17–62) years | 24 | Fibula = 11 DCIA = 13 | NR | 1 | 1–12 months | general morbidity and quality of life; functional and aesthetic evaluation of the donor site and oromandibular complex. | High |
| Wilkman, et al. 2018 Finland | RS | (median, 61 years; range, 18–89 years) | 163 patients | Fibula = 18 Scapula = 32 DCIA = 72 | NR | 8 | NR | Duration of surgery; blood loss; dental implant; ICU stay; hospital stay; dental implant | Low |
| Winters et al.2006 Netherland | RS | 54 years (range 11 to 80) | 72 | Fibula = 45 DCIA = 27 | NR | 3 | NR | Revision Surgery, Recipient site morbidity, Donor site morbidity | Moderate |

RC = retrospective study; NR = not reported; RCT = randomized controlled trial.

(*) Jewer's mandibular classification.

(#) Schultz et al mandibular defect classification.

(**) lateral defect with a soft tissue resection limited to the oral cavity and/or oropharynx; type 2 lateral defect with a through and through defect of the lower one-third of the face (skin overlying the mandible) or neck; and type 3 lateral defect with an associated large volume resection of the midface, parotid, and/or cheek skin.

(#) 15 patients reconstructed with cutaneous radial forearm flap were excluded.

method in gemtc package in R [26] (R Foundation for Statistical Computing, Vienna, Austria); GeMTC employs JAGS software-based comparative calculations.

## 3. Results

The initial electronic and manual searches yielded 421 potentially relevant articles (Fig 1). Of them, 384 studies remained after removal of duplication. The titles and abstracts of the remaining 384 articles were screened, and then further 339 studies were excluded due to being off-topic or non-English studies. Two researchers carefully read the full texts of the remaining 45 studies for potential inclusion. There were 22 studies [27–47] fulfilling the inclusion criteria (Table 1). The other 23 studies were excluded with reasons (S4 Table). There were one RCT [33] and 21 non-RCTs [27–32, 34–48]. The included studies were published between 1985 and 2020 and compared four common vascularized osseous flaps. A total of 1513 patients underwent mandibular reconstruction and received one of the following vascularized osseous flaps: FFF, DCIA, Scapula flap, and ORFF. The follow-up of the included studies ranged from 0.5 month to 7 years (Table 1).

### 3.1. Study characteristics

There were 22 articles, the respective survival rate of FFF (n = 671), DCIA (n = 539), Scapula (n = 105), and ORFF (n = 198) was 94.50%, 93.12%, 97%, and 95.95%. Two studies [31, 41] were excluded because they reported 0 event in both interventions. Four common vascularized bony flaps for mandibular reconstruction were reported in the current NMA. Other general characteristics of the included studies are summarized in Table 1.

### 3.2. Risk of bias assessment

A summary of the risk of bias assessment is shown in the S3 Table and Table 1. One RCT was assessed as having a high risk of bias. Regarding NOS for the non-RCTs, there were 11 studies assessed as having low risk and 10 studies were assessed as having moderate risk of bias (S3 Table and Table 1).

### 3.3. Outcomes of pairwise meta-analysis

Fourteen studies [27, 28, 30, 32–36, 38–40, 45, 46, 49] with a total of 761 (FFF = 409 and DCIA = 352) patients compared FFF versus DCIA. Two studies [31, 41] were excluded from the meta-analysis because they reported 0 flap failure in both groups. The results of the pairwise meta-analysis failed to reveal a statistically significant difference in flap survival rate (Fixed, Odd ratio 0.901; 95% CI, 0.475; 1.708, P = 0.749, I2 = 0%) (Fig 2).

FFF was compared with ORFF in 3 studies [37, 43, 47] with a total of 331 patients. FFF failed to show a statistically significant difference when compared with ORFF (Fixed, Odd ratio, 1.160; 95% CI, 0.3665; 3.670, P = 0.801, I2 = 0%) (S1 Fig).FFF were compared with the scapula flap in 3 studies [27, 32, 42] with a total of 248 (FFF = 110 and Scapula = 138) patients. FFF failed to show a statistically significant difference compared to scapula flap regarding flap survival rate (Fixed, Odd ratio, 2.774; 95% CI, 0.707; 10.876, P = 0.143, I2 = 8%) (S1 Fig).

DCIA flap was compared with the scapula flap in 2 studies [27, 32] with a total of 191 (DCIA = 108 and Scapula = 83) patients. Scapula flap showed a statistically significant difference compared to DCIA regarding flap survival rate (Fixed, Odd ratio 5.436; 95% CI, 1.285; 22.995, P = 0.021, I2 = 0%) (S1 Fig).

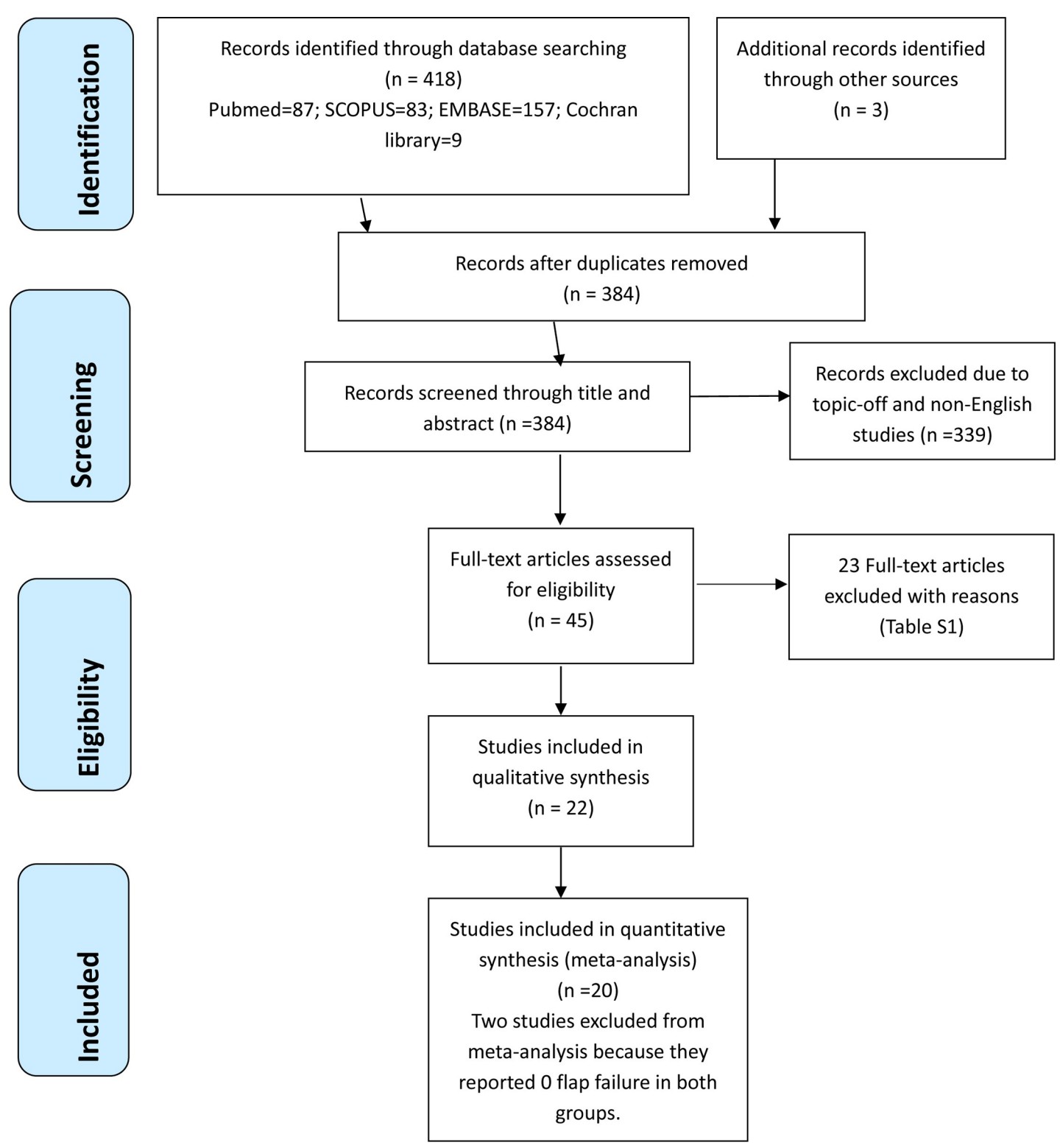

**Fig 1. Study flow diagram.**

# Meta Analysis

| Study name | Statistics for each study | | | | | Odds ratio and 95% CI |
|---|---|---|---|---|---|---|
| | Odds ratio | Lower limit | Upper limit | Z-Value | p-Value | |
| Haughey et al. 1994 | 8.333 | 0.351 | 198.086 | 1.312 | 0.190 | |
| Heller et al. 1995 | 11.000 | 0.380 | 318.606 | 1.396 | 0.163 | |
| Shpitzer et al. 1985 | 1.761 | 0.401 | 7.735 | 0.749 | 0.454 | |
| Takushima et al. 2001 | 0.862 | 0.237 | 3.138 | -0.225 | 0.822 | |
| Chang et al. 2001 | 0.094 | 0.006 | 1.393 | -1.719 | 0.086 | |
| Yilmaz et al. 2008 | 0.580 | 0.022 | 15.265 | -0.326 | 0.744 | |
| Van Germert et al. 2011 | 0.088 | 0.004 | 1.919 | -1.545 | 0.122 | |
| Hanken et al. 2014 | 0.673 | 0.024 | 18.845 | -0.233 | 0.816 | |
| T. Wilkman, et al. 2018 | 0.546 | 0.063 | 4.747 | -0.548 | 0.584 | |
| Ritschl et al. 2020 | 2.579 | 0.134 | 49.573 | 0.628 | 0.530 | |
| Schultz et al. 2015 | 0.892 | 0.032 | 25.147 | -0.067 | 0.946 | |
| Chen et al. 1994 | 0.512 | 0.020 | 13.191 | -0.404 | 0.686 | |
| Politi,and Toro. 2012 | 3.857 | 0.142 | 104.645 | 0.802 | 0.423 | |
| | 0.980 | 0.505 | 1.902 | -0.060 | 0.953 | |

Favours A    Favours B

**Meta Analysis**

**Fig 2. Forest plot pairwise direct comparison of FFF versus DCIA.**

DCIA flap was compared with ORFF in one study [44] with a total of 73 (DCIA = 60 and ORFF = 13) patients and no statistically significant difference was observed regarding flap survival rate. (S1 Fig).

## 3.4. Outcomes of network meta-analysis

Flap survival rate was reported in 22 studies, two studies [31, 41] reported 0 events in both groups and therefore were excluded from NMA. The network diagram of the 20 included studies concerning flap survival rate with different osseous flaps for mandibular reconstruction is presented in Fig 3. The results of the NMA of the 20 included studies are presented in Fig 4. There were statistically insignificant differences between all comparators as shown in Fig 4.

The results of NMA failed to show a statistically significant difference when FFF was compared with DCIA (Odd ratio, 1.8; CrI, 0.58,5.0); ORFF (Odd ratio, 0.57; CrI, 0.077; 2.9) and scapula flap (Odd ratio, 0.25; CrI, 0.026; 1.5). Also, DCIA failed to show a statistically significant difference when compared with ORFF (Odd ratio, 0.32; CrI, 0.037; 2.1) and scapula flap (Odd ratio, 0.14; CrI, 0.015; 1.1). Similarly, no significant difference was observed when ORFF was compared with the scapula flap (Odd ratio, 2.3; CrI, 0.16; 34) (Fig 4).

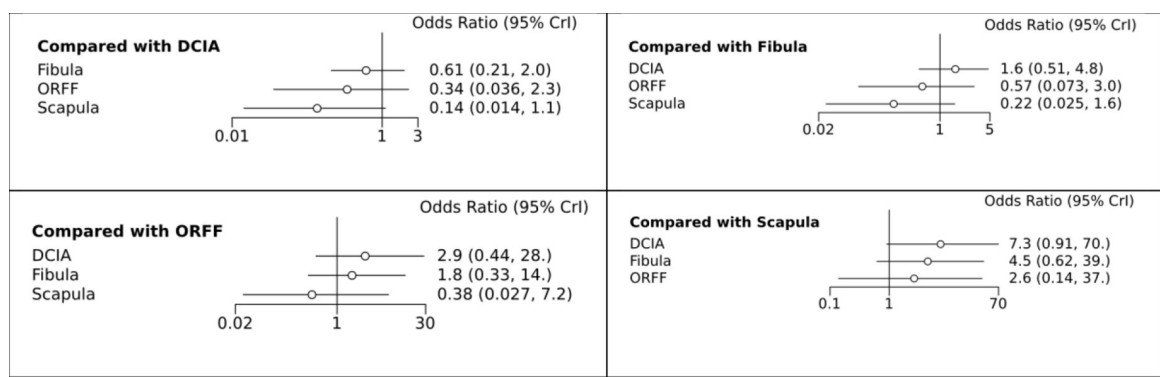

**Fig 3. Network map of all direct comparisons of reported by all included articles in the network meta-analysis.** The thickness of the line correlates with the number of comparisons made.

**Compared with DCIA**

| | Odds Ratio (95% CrI) |
|---|---|
| Fibula | 0.61 (0.21, 2.0) |
| ORFF | 0.34 (0.036, 2.3) |
| Scapula | 0.14 (0.014, 1.1) |

0.01    1  3

**Compared with Fibula**

| | Odds Ratio (95% CrI) |
|---|---|
| DCIA | 1.6 (0.51, 4.8) |
| ORFF | 0.57 (0.073, 3.0) |
| Scapula | 0.22 (0.025, 1.6) |

0.02    1    5

**Compared with ORFF**

| | Odds Ratio (95% CrI) |
|---|---|
| DCIA | 2.9 (0.44, 28.) |
| Fibula | 1.8 (0.33, 14.) |
| Scapula | 0.38 (0.027, 7.2) |

0.02    1    30

**Compared with Scapula**

| | Odds Ratio (95% CrI) |
|---|---|
| DCIA | 7.3 (0.91, 70.) |
| Fibula | 4.5 (0.62, 39.) |
| ORFF | 2.6 (0.14, 37.) |

0.1   1    70

**Fig 4. NMA forest plot, comparison of the included interventions: Odds ratio (95% CrI).**

### 3.5. Rank probabilities plot

The rank probabilities plot concerning the flap survival rate is presented in Fig 5 the rank probability test showed that the scapula flap ranked the best which gained 73.4% of all treatments and was associated with the least incidence of flap failure; followed by ORFF (24.6%), FFF (1.6%), and DCIA (0.4%) (Fig 5).

### 3.6. Node splitting and publication bias

The results of the node-splitting analysis along with its inconsistency P-value showed insignificant differences between direct and indirect evidences (P>0.05) (S2 Fig), indicating that the results of the current NMA are reliable. Comparison-adjusted funnel plots were constructed

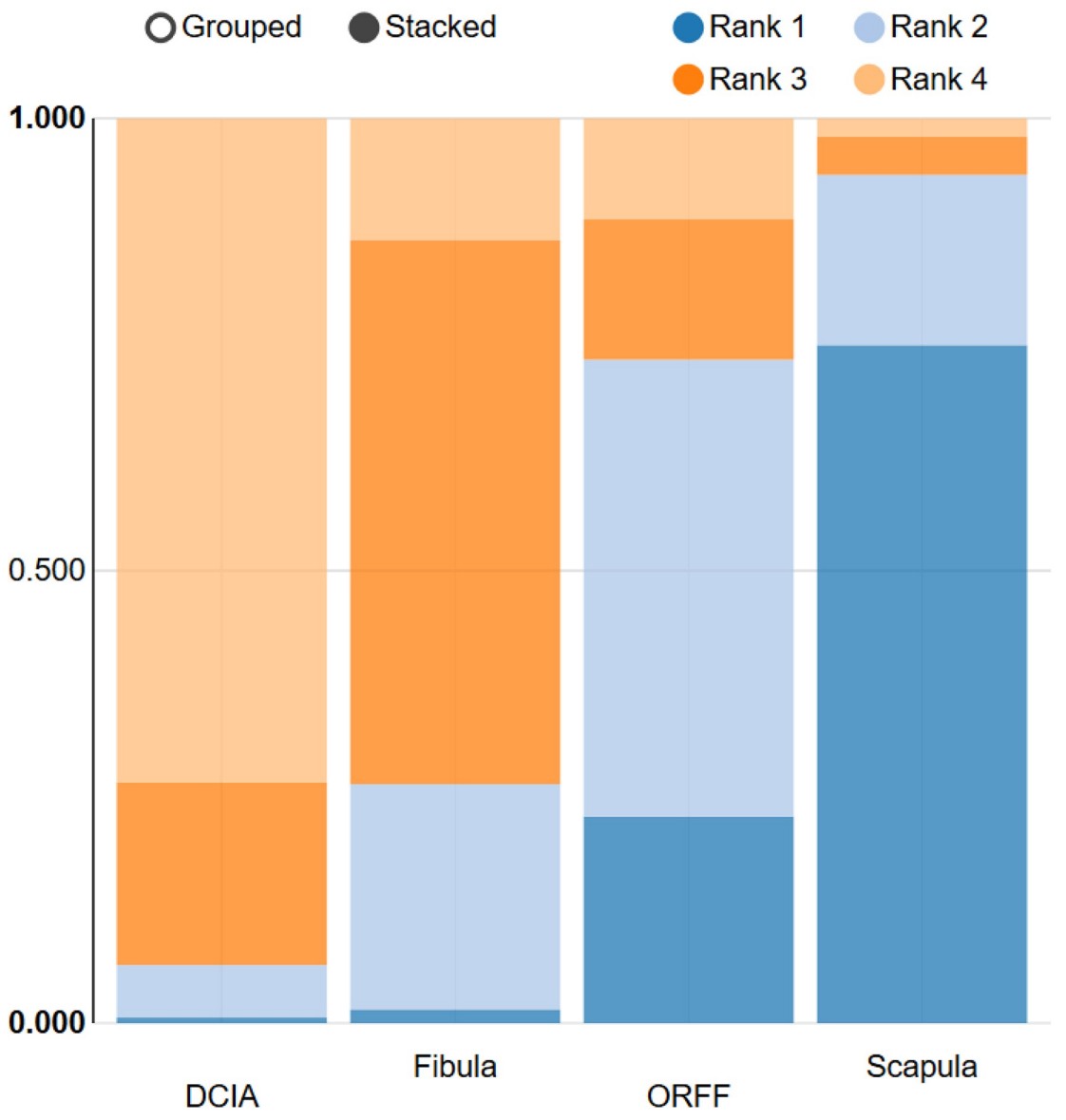

**Fig 5. Rankogram of different osseous flap.** Rank 1 correlates with the lowest incidence of flap failure.

for each osseous flap, which showed an even distribution of studies adjacent to the pooled estimate line, indicating the absence of a small size effect and publication bias (Fig 6).

## 4. Discussion

Unlike the pair wise meta-analysis that can provide a direct comparison of only two interventions, network meta-analysis is valuable tool for clinical decision-making because it allows

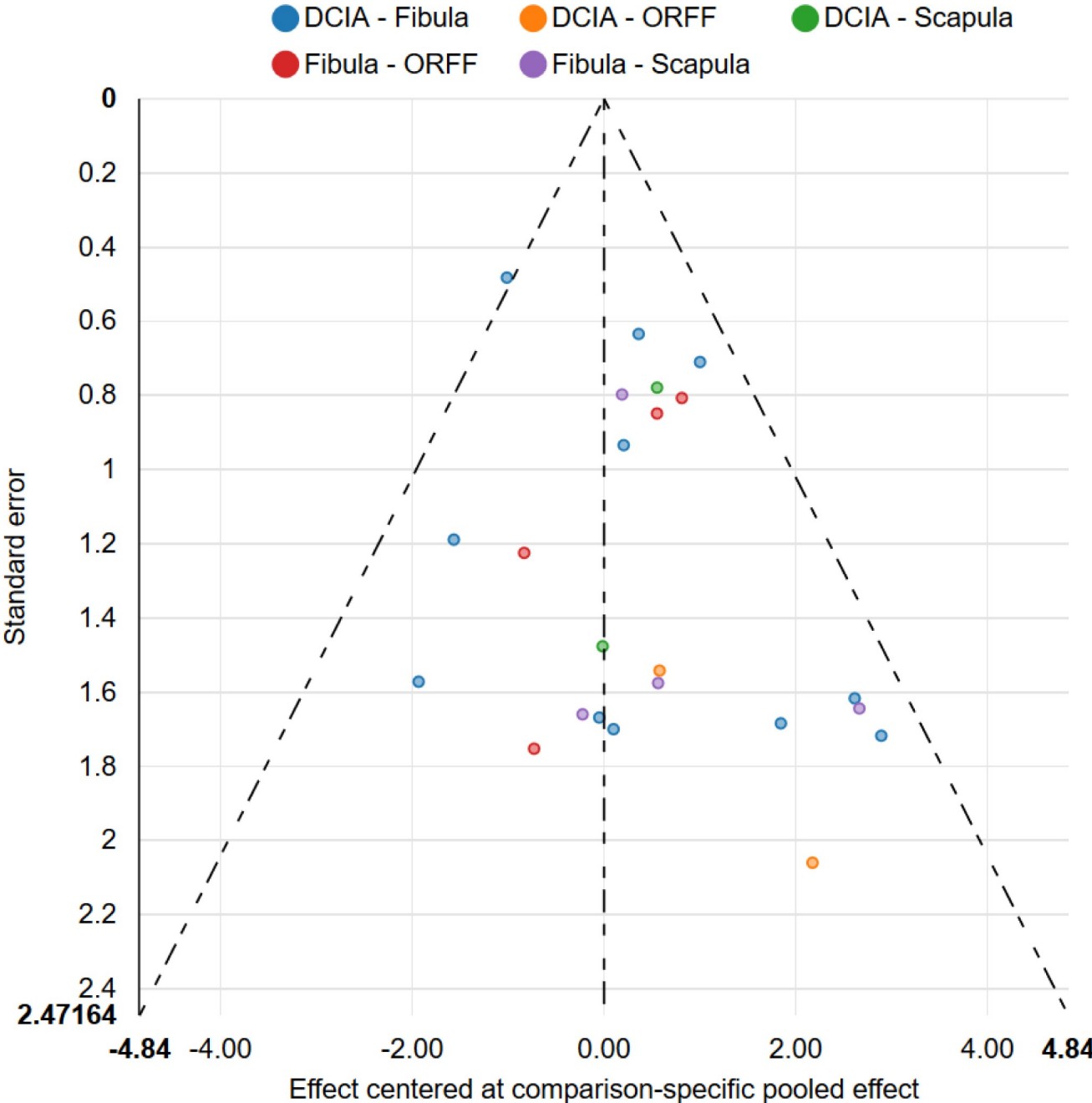

**Fig 6. Comparison-adjusted funnel plots of different osseous free flaps.**

comparisons of multiple interventions and providing an evidence regarding which intervention is the best, In addition, it can provide indirect evidence by comparing of multiple interventions that may not have been studied in a head-to-head fashion, thus potentially facilitating timely recommendations and reducing research waste [50]. The current NMA represents several advantages as follows: First, it advocates the concept of considering other factors than only flap survival in selecting an osseous flap for mandibular reconstruction. Second, it poses an emphasis on the relevant clinical evidence available in the literature for survival analysis following osseous flaps.

To the best of the authors' knowledge, this is the first NMA to summarize the available evidence regarding which vascularized osseous flap is associated with the highest survival rate. The present NMA included 20 studies compared four osseous flaps namely FFF, DCIA, Scapula, and ORFF for mandibular reconstruction. The result of NMA failed to show a statistically significant difference in flap survival rate between FFF, DCIA, scapula, and ORFF. This was in line with the results of the conventional meta-analysis conducted by Markiewicz et al [19] and Lonie et al [20] in that no difference was observed between FFF and DCIA regarding flap survival rate. Wilkman et al [32] compared DCIA with Scapula and FFF and they reported a statistically significant difference in the flap survival rate favouring Scapula flap and this was inconsistent with the results of the current NMA. Van Genechten et al [51] showed no difference in the survival rate of FFF, DCIA, ORFF, and Scapula flap which is consistent with the current study.

Mucke et al [52] considered the type of flap as one of the risk factors for flap loss and they stated that FFF was significantly associated with less flap loss compared with DCIA. Militsakh et al [53] compared ORFF with FFF and scapula flap and they found no difference in the flap survival rate and this was consistent with the results of the current pairwise and network meta-analysis.

Ideally, an osseous flap used for reconstruction of oromandibular defect should be reliable, functional, cosmetically acceptable, resemble the mandible in terms of width, length, and thickness, having sufficient bone for dental implant placement and associated with minimal donor site morbidity.

The evolution of microsurgery allows the application of various vascularized bone flaps to restore mandibular continuity, however, dental implants are the only available option for optimal restoring of both aesthetics and functions. Although the free vascularized fibula flap is considered as the workhorse flap for functional mandible reconstruction, however, still its main drawback is the discrepancy of the vertical height as the transplanted fibula bone is often shorter than the native mandible. Accordingly, there would be difficulty in placement of dental implants or even while using the ordinary dentures. Furthermore, the vertical height discrepancy results in various disadvantages including changing the normal facial contour, in addition to placement of dental implants with long clinical crowns. Management of deficient vertical height includes different alternatives; fixation of the transplanted fibula superior to the lower border of mandible, vertical distraction osteogenesis of the fibula bone, the double-barrel vascularized fibula technique, and the onlay bone grafts [54]. However, considering each individual method, still some limitations exist. Of all, the outcome of the double barrel fibula flap is the most reliable and predictable. Unfortunately, this technique is limited to mandibular defects not more than 8cm to have a dentally rehabilitated patient with perfect functional and aesthetic outcomes [17, 54].

Contrary, the DCIA flap offers a distinct advantage over the single barrel fibular flap, in that it provides adequate bone quality and quantity for placement of dental implants that mimics the vertical height of the native mandible. Most importantly, the curved shape of the iliac crest that suits the parabolic shape of the mandible especially the angle-body region [18].

Considering the survival rates of osseointegrated implants among various vascularized osseous flaps, Laverty et al reported 100% survival of 12 implants in scapula free flap, 83.1% survival of 64 implants in FFF, 76% survival of 25 implants in DCIA and 80% survival of 15 implants in RFFF [55]. However, in Systematic review conducted by Khadembaschi et al, there has been acceptable results with no differences between FFF and DCIA. As such, clinical decisions for use of implants should be made on patient and deficit factors [56].

Recently, the scapula flap has gained wide popularity for bony reconstruction of the head and neck region. It has many advantages making it an ideal osseous flap for replacement of composite tissue defects, such as adequate bone stock for the reconstruction of lateral mandibular defects, great mobility of the skin paddle in relation to bone, the option of simultaneous tumor resection and flap harvest is often possible [57], can be harvested as a chimeric flap (scapula-latissimus flap) in cases in which massive multiplanes soft tissue defects exist, has an adequate pedicle length and large vessels caliber and lack any atherosclerotic disease. However, donor site morbidities such as shoulder dysfunction and brachial plexus injury are the main complications that may result after scapula flap harvest. Also, it cannot be used for total mandibular reconstruction due to limited bone stock. Furthermore, the proximity of the resection and reconstruction teams may lead to crowding and prolonging the operative time.

Interestingly, the result of the rank probability test showed that the scapula flap is associated with the highest survival rate for mandibular reconstruction and this was in line with several studies that reported 100% survival rate [32, 58]. Also, this result is consistent with the study that confirmed the superiority of scapula flap compared to FFF and DCIA [32].

Radial forearm flap has been widely used for reconstruction of intra-oral defects. It has many advantages making it a favorable flap for tissue replacement such as a long vascular pedicle, constant, reliable anatomy, ease of harvest, thin pliable, supple skin, two-teams approach, and a high success rate [59]. However, donor site complications continue to be one of its major disadvantages [59, 60], particularly, pathological donor radius fracture when raised as an osseous flap [61]. In a systematic review conducted by Kearns et al [62], the authors concluded that the scapula flap is associated with the lowest donor site morbidity and should be strongly considered when the recipient defect allows whereas the ORFF is associated with higher morbidity and should not be considered as the first choice when other flap options are available.

FFF and DCIA were compared in 14 studies with a total of 761 patients, and a statistically insignificant difference was reported in both pairwise and network meta-analysis. FFF and DCIA have proved their usefulness in the reconstruction of mandibular defects after ablative surgery. Both flaps share many ideal advantages for bony replacement after oncological mandibulectomy. For instance, both have long pedicle length, adequate vessel diameter, ability to incorporate skin paddle and muscle, the simultaneous two-team approach is also possible, and provide sufficient bone height and width for dental implant placement. However, FFF is contraindicated in patients with peripheral vascular disease and those with lower limb comorbidities such as arthritis. Furthermore, delayed mobilization, especially in elderly patients is considered to be one of its major downsides [29]. On the other hand, DCIA has been criticized for the limited mobility of the skin paddle in relation to bone and the limited bone stock making it insufficient for total mandibular reconstruction.

The results of the current NMA suggests that flap survival should not be considered as an important factor in decision making when choosing a donor site for mandibular reconstruction. Alternatively, deciding which osseous flap best fits the patient's needs to be customized based on many factors, of them, subjective own preferences of the surgeon, patient's general condition, operative time, donor site morbidity, bone stock, defect size, skin paddle characteristics, pedicle length, and the intention for dental implant rehabilitation [63].

Although the current NMA is the first and largest study evaluating different osseous flaps for mandibular reconstruction; however, some limitations in this study have to be declared. First, with the exception of only one RCT, other studies were non-RCT. However, conducting a comparative, carefully designed RCT in the field of surgery appears to be difficult. Second, most of the included studies are rated as having a high risk of bias. Third, most of the included studies evaluated mandibular reconstruction after surgical resection of both malignant and benign tumours. So, it is still unclear whether the tumour entity affects flap survival or not.

Forth, several cofounders such as the type of mandibular defect, patient's comorbidity (i.e, severe peripheral vascular disease), length of the osseous bone, number of osteotomies, the timing of dental implant placement, surgeon experience, etc, that might play a role in flap survival, were not reported in most of the included studies. Fifth, there were no enough studies to compare different osseous flaps regarding other complications (systemic complication, recipient site, or donor site complications). Nonetheless, the current NMA answered the question regarding which osseous flap is associated with the highest survival rate. Therefore, future multicentre RCTs with longer follow-up and larger sample size comparing the different osseous flaps in regards to other complications are still needed.

In conclusion, the current NMA demonstrated that the FFF, DCIA, Scapula flap, and ORFF are reliable and showed no differences regarding flap survival rate when used in the reconstruction of mandibular defects. Although the scapula flap reported the highest survival rate, however, the decision regarding which osseous flap to be used for mandibular reconstruction should be based on factors other than flap survival rate such as surgeon preference, patient's age, patient's medical condition, defect size, the extent and location of defects, and the intention for dental implant rehabilitation.

## Supporting information

**S1 Fig. Forest plot result of a pairwise meta-analysis of direct comparison of the different osseous free flap for mandibular reconstruction.**
(PDF)

**S2 Fig. Results of node splitting concerning flap survival rate along with inconsistency Bayesian P-value.**
(PDF)

**S1 Table. PRISMA checklist.**
(DOCX)

**S2 Table. Search strategy terms.**
(DOCX)

**S3 Table. Risk of bias assessment.**
(DOCX)

**S4 Table. Excluded studies and reasons for exclusion.**
(DOCX)

## Author Contributions

**Conceptualization:** Mubarak Ahmed Mashrah.

**Data curation:** Mubarak Ahmed Mashrah, Taghrid Aldhohrah.

**Formal analysis:** Mubarak Ahmed Mashrah, Taghrid Aldhohrah.

**Funding acquisition:** Liping Wang.

**Investigation:** Mubarak Ahmed Mashrah, Hamada Mahran, Faisal Abu-lohom, Hanfu Su.

**Methodology:** Mubarak Ahmed Mashrah, Taghrid Aldhohrah, Karim Ahmed Sakran, Hyat Ahmad, Hamada Mahran, Faisal Abu-lohom, Hanfu Su.

**Project administration:** Mubarak Ahmed Mashrah.

**Resources:** Mubarak Ahmed Mashrah.

**Software:** Mubarak Ahmed Mashrah.

**Supervision:** Faisal Abu-lohom, Ying Fang, Liping Wang.

**Validation:** Mubarak Ahmed Mashrah, Karim Ahmed Sakran, Faisal Abu-lohom.

**Visualization:** Mubarak Ahmed Mashrah, Hamada Mahran.

**Writing – original draft:** Mubarak Ahmed Mashrah.

**Writing – review & editing:** Mubarak Ahmed Mashrah, Ahmed Abdelrehem.

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
