## [Decision Letter · Decision Letter 0]

17 Aug 2021

PONE-D-21-16832

Survival of vascularized osseous flaps in mandibular reconstruction: A network meta-analysis

PLOS ONE

Dear Dr. Mashrah,

Thank you for submitting your manuscript to PLOS ONE. After careful consideration, we feel that it has merit but does not fully meet PLOS ONE’s publication criteria as it currently stands. Therefore, we invite you to submit a revised version of the manuscript that addresses the points raised during the review process.

ACADEMIC EDITOR: MAJOR REVISION

I agree with the reviewers. Please address all of their comments and questions

We look forward to receiving your revised manuscript.

Kind regards,

Peter Dziegielewski, MD, FRCSC

Academic Editor

PLOS ONE

1. Please ensure that your manuscript meets PLOS ONE's style requirements, including those for file naming. The PLOS ONE style templates can be found at https://journals.plos.org/plosone/s/file?id=wjVg/PLOSOne_formatting_sample_main_body.pdf and https://journals.plos.org/plosone/s/file?id=ba62/PLOSOne_formatting_sample_title_authors_affiliations.pdf.

2. In the Methods section, please report the date range over which the literature search was conducted.

“No”

Please include your amended statements within your cover letter; we will change the online submission form on your behalf.4. We note that this manuscript is a systematic review or meta-analysis; our author guidelines therefore require that you use PRISMA guidance to help improve reporting quality of this type of study. Please upload copies of the completed PRISMA checklist as Supporting Information with a file name “PRISMA checklist”.

Additional Editor Comments (if provided):

Thank you for your submission. I agree with the reviewers. Some edits/revision are needed.

Reviewers' comments:

Reviewer's Responses to Questions

**Comments to the Author**

1. Is the manuscript technically sound, and do the data support the conclusions?

Reviewer #1: Yes

Reviewer #2: Yes

2. Has the statistical analysis been performed appropriately and rigorously? 

Reviewer #1: Yes

Reviewer #2: Yes

3. Have the authors made all data underlying the findings in their manuscript fully available?

Reviewer #1: Yes

Reviewer #2: Yes

4. Is the manuscript presented in an intelligible fashion and written in standard English?

Reviewer #1: Yes

Reviewer #2: Yes

5. Review Comments to the Author

Reviewer #1: Strong work on this paper. The methodology used to construct the network meta-analysis is sound. The results and conclusion of the study work to support other existing literature including more traditional meta-analyses. I think it is in the authors' interests to further explain the uniqueness of their work being a network meta-analysis and perhaps explain how this differs from other forms of meta-analysis and how this can bolster the existing literature on free flap reconstruction of mandibular defects. I recommend a thorough editing of the submitted manuscript for spelling and grammar, as there are a number of mistakes in this area. I would caution the authors to avoid using the phrase 'statistically insignificant' and rather state that the finding 'failed to show a statistically significant difference' or was 'not statistically significant' to avoid any confusion and in-line with norms in scientific literature.

Reviewer #2: I believe this article to be very well written and I enjoyed the meta analysis and rigor dedicate to uncovering bias in this paper. I believe the results to be valid. I do not think the conclusions offer any groundbreaking information that was previously not available as these flaps have been used for decades and those flap surgeons continuing to use the selected flaps have mastered their success to be in the mid to high 90% tile.

This being said I believe the value in this article is a paper that can place concise direct comparisons between the groups even though showing negative results.

One area I feel minor revisions could be added in the discussion is another paragraph dedicated to dental implantation. This has become the standard and is often desired by patients undergoing mandibular reconstruction. The paper does briefly touch on this in the discussion, however I feel more discussion is worthy given the success of all flaps essentially equal this will usually be a deciding factor on which flap the Head and Neck surgeon will employ, providing no other patient specific factors (ex. severe peripheral vascular disease).

6. PLOS authors have the option to publish the peer review history of their article (what does this mean?). If published, this will include your full peer review and any attached files.

Reviewer #1: **Yes: **Robert M. Liebman

Reviewer #2: No

---

## [Author Response · Author response to Decision Letter 0]

23 Aug 2021

Response to the reviewers’ comments

PONE-D-21-16832

Survival of vascularized osseous flaps in mandibular reconstruction: A network meta-analysis

PLOS ONE

Dear Prof. Dr. Peter Dziegielewski, MD, FRCSC

Academic Editor

PLOS ONE

Thank you very much for your time and effort spent on reviewing our manuscript and for your kind help. We would like to thank all the reviewers for their thorough and thoughtful comments. We have revised our manuscript based on a point-by-point response to all comments and all the changes have been highlighted in red in the text. Meanwhile, we have done some explanation of our work according to your comments point by point as follows. We hope our manuscript will be suitable for publication in PLOS ONE.

Once again, we greatly appreciate all the efforts done by the editor and reviewers for allowing us to further improve our manuscript.

*Corresponding author:

Mubarak Ahmed Mashrah, Ying Fang and Wang Liping 

Key Laboratory of Oral Medicine, Guangzhou Institute of Oral Disease, Stomatology Hospital of Guangzhou Medical University, Guangdong Guangzhou 510140, China

mubarak198226@gmail.com; and wangliplj@126.com.

Mobile: 0086-17620963636

Dear Dr. Mashrah,

Thank you for submitting your manuscript to PLOS ONE. After careful consideration, we feel that it has merit but does not fully meet PLOS ONE’s publication criteria as it currently stands. Therefore, we invite you to submit a revised version of the manuscript that addresses the points raised during the review process.

ACADEMIC EDITOR: MAJOR REVISION

I agree with the reviewers. Please address all of their comments and questions

We look forward to receiving your revised manuscript.

Kind regards,

Peter Dziegielewski, MD, FRCSC

Academic Editor

PLOS ONE

Our response to the academic editor:

Thank you very much for your time and effort spent on reviewing our manuscript and for your kind help. All the comments raised by you and the reviewers are now addressed as suggested. 

1. Please ensure that your manuscript meets PLOS ONE's style requirements, including those for file naming. The PLOS ONE style templates can be found at https://journals.plos.org/plosone/s/file?id=wjVg/PLOSOne_formatting_sample_main_body.pdf and https://journals.plos.org/plosone/s/file?id=ba62/PLOSOne_formatting_sample_title_authors_affiliations.pdf.

2. In the Methods section, please report the date range over which the literature search was conducted.

Our response: 

Many thanks for your valuable comments! the date range over which the literature search is now provided (please see lines 45 and 137)

“No”

Please include your amended statements within your cover letter; we will change the online submission form on your behalf.4. We note that this manuscript is a systematic review or meta-analysis; our author guidelines therefore require that you use PRISMA guidance to help improve reporting quality of this type of study. Please upload copies of the completed PRISMA checklist as Supporting Information with a file name “PRISMA checklist”. Additional Editor Comments (if provided):

Thank you for your submission. I agree with the reviewers. Some edits/revision are needed.

Our response: 

Many thanks for your valuable comments! We confirm that the manuscript meets PLOS ONE's style requirements. All the items mentioned above are now addressed as suggested. 

Reviewers' comments:

Reviewer's Responses to Questions

Comments to the Author

1. Is the manuscript technically sound, and do the data support the conclusions?

Reviewer #1: Yes

Reviewer #2: Yes

Our response: 

Many thanks for your positive feedback!

 2. Has the statistical analysis been performed appropriately and rigorously? 

 Reviewer #1: Yes

Reviewer #2: Yes

Our response: 

Many thanks for your positive feedback!

3. Have the authors made all data underlying the findings in their manuscript fully available?

 Reviewer #1: Yes

Reviewer #2: Yes

Our response: 

Many thanks for your positive feedback! 

4. Is the manuscript presented in an intelligible fashion and written in standard English?

 Reviewer #1: Yes

Reviewer #2: Yes

Our response: 

Many thanks for your positive feedback! 

5. Review Comments to the Author

Reviewer #1: 

Strong work on this paper. The methodology used to construct the network meta-analysis is sound. The results and conclusion of the study work to support other existing literature including more traditional meta-analyses. I think it is in the authors' interests to further explain the uniqueness of their work being a network meta-analysis and perhaps explain how this differs from other forms of meta-analysis and how this can bolster the existing literature on free flap reconstruction of mandibular defects. 

Our response: 

Many thanks for your positive feedback! Some systematic reviews compare only two interventions, in which a conventional pair-wise meta-analysis may be conducted, while others examine the comparative effectiveness of many or all available interventions for a given condition. When the comparative effectiveness of a range of interventions is of interest, appropriate statistical methodology must be used for analysis. Network meta-analysis expands the scope of a conventional pair-wise meta-analysis by comparing multiple treatments and analyzing simultaneously both direct comparisons of interventions and indirect comparisons across trials based on a common comparator. In the simplest case, one may be interested in comparing two interventions A and C. Indirect evidence can be obtained from study of either A or C versus a common comparator B. When both direct and indirect evidence are available, the two sources of information can be combined as a weighted average when appropriate. Using statistical methods to combine findings from individual studies in network meta-analysis can provide evidence for selecting the best treatment from several treatment options. In a ward, Network meta-analysis holds promise to provide evidence on comparative effectiveness that is valuable for clinical decision-making because it allows comparisons of interventions that may not have been directly compared in head-to-head trials, thus potentially facilitating timely recommendations and reducing research waste. (This has been added in the introduction part in the first submission).

We added the following paragraph in the discussion part (please see lines 311-321)

Unlike pair wise meta-analysis that can provide a direct comparison of only two interventions, network meta-analysis is valuable tool for clinical decision-making because it allows comparisons of multiple interventions and providing an evidence regarding which intervention is the best, In addition, it can provide indirect evidence by comparing of multiple interventions that may not have been studied in a head-to-head fashion, thus potentially facilitating timely recommendations and reducing research waste [50]. The current NMA represents several advantages as follows: First, it advocates the concept of considering other factors than only flap survival in selecting an osseous flap for mandibular reconstruction. Second, it poses an emphasis on the relevant clinical evidence available in the literature for survival analysis following osseous flaps.

I recommend a thorough editing of the submitted manuscript for spelling and grammar, as there are a number of mistakes in this area.

Our response: 

Many thanks for your valuable comment! The spilling and grammar mistakes are now edited by English native speaker.

 I would caution the authors to avoid using the phrase 'statistically insignificant' and rather state that the finding 'failed to show a statistically significant difference' or was 'not statistically significant' to avoid any confusion and in-line with norms in scientific literature.

Our response: 

Many thanks for your valuable comment! The phrase “'statistically insignificant'” are now replaced with 'failed to show a statistically significant difference'.

Reviewer #2: I believe this article to be very well written and I enjoyed the meta analysis and rigor dedicate to uncovering bias in this paper. I believe the results to be valid. I do not think the conclusions offer any groundbreaking information that was previously not available as these flaps have been used for decades and those flap surgeons continuing to use the selected flaps have mastered their success to be in the mid to high 90% tile.

This being said I believe the value in this article is a paper that can place concise direct comparisons between the groups even though showing negative results. One area I feel minor revisions could be added in the discussion is another paragraph dedicated to dental implantation. This has become the standard and is often desired by patients undergoing mandibular reconstruction. The paper does briefly touch on this in the discussion, however I feel more discussion is worthy given the success of all flaps essentially equal this will usually be a deciding factor on which flap the Head and Neck surgeon will employ, providing no other patient specific factors (ex. severe peripheral vascular disease).

Our response: 

Many thanks for your positive feedback and valuable comment! 

One paragraph about dental implant is now added in the discussion part (Please see lines 344-375 ). 

The following paragraph is now added: The evolution of microsurgery allows the application of various vascularized bone flaps to restore mandibular continuity, however, dental implants are the only available option for optimal restoring of both aesthetics and functions. Although the free vascularized fibula flap is considered as the workhorse flap for functional mandible reconstruction, however, still its main drawback is the discrepancy of the vertical height as the transplanted fibula bone is often shorter than the native mandible. Accordingly, there would be difficulty in placement of dental implants or even while using the ordinary dentures. Furthermore, the vertical height discrepancy results in various disadvantages including changing the normal facial contour, in addition to placement of dental implants with long clinical crowns. Management of deficient vertical height includes different alternatives; fixation of the transplanted fibula superior to the lower border of mandible, vertical distraction osteogenesis of the fibula bone, the double-barrel vascularized fibula technique, and the onlay bone grafts[54]. However, considering each individual method, still some limitations exist. Of all, the outcome of the double barrel fibula flap is the most reliable and predictable. Unfortunately, this technique is limited to mandibular defects not more than 8cm to have a dentally rehabilitated patient with perfect functional and aesthetic outcomes [17][54].

 Contrary, the DCIA flap offers a distinct advantage over the single barrel fibular flap, in that it provides adequate bone quality and quantty for placement of dental implants that mimics the vertical height of the native mandible. Most importantly, the curved shape of the iliac crest that suits the parabolic shape of the mandible especially the angle-body region [18]. 

Considering the survival rates of osseointegrated implants among various vascularized osseous flaps, Laverty et al reported 100% survival of 12 implants in scapula free flap, 83.1% survival of 64 implants in FFF, 76% survival of 25 implants in DCIA and 80% survival of 15 implants in RFFF [55]. However, in Systematic review conducted by Khadembaschi et al, there has been acceptable results with no differences between FFF and DCIA. As such, clinical decisions for use of implants should be made on patient and deficit factors [56]. 

In addition, we agree with the reviewer in that” the success of all flaps essentially equal this will usually be a deciding factor on which flap the Head and Neck surgeon will employ, providing no other patient specific factors (ex. severe peripheral vascular disease).” 

The current study also failed to show a statistically significant difference between osseous flaps concerning flap survival rate. This was added as one of the limitation of the current study, (Please see line 436-440) several cofounders such as the type of mandibular defect, patient’s comorbidity (i.e, severe peripheral vascular disease), length of the osseous bone, number of osteotomies, the timing of dental implant placement, surgeon experience, etc, that might play a role in flap survival, were not reported in most of the included studies.

6. PLOS authors have the option to publish the peer review history of their article (what does this mean?). If published, this will include your full peer review and any attached files.

Do you want your identity to be public for this peer review? For information about this choice, including consent withdrawal, please see our Privacy Policy.

 Reviewer #1: Yes: Robert M. Liebman

Reviewer #2: No

 While revising your submission, please upload your figure files to the Preflight Analysis and Conversion Engine (PACE) digital diagnostic tool, https://pacev2.apexcovantage.com/. PACE helps ensure that figures meet PLOS requirements. To use PACE, you must first register as a user. Registration is free. Then, login and navigate to the UPLOAD tab, where you will find detailed instructions on how to use the tool. If you encounter any issues or have any questions when using PACE, please email PLOS at figures@plos.org. Please note that Supporting Information files do not need this step.________________________________________In compliance with data protection regulations, you may request that we remove your personal registration details at any time. (Remove my information/details). Please contact the publication office if you have any questions.

We carefully considered the comments offered by the referees. We revised the paper according to the reviewers’ comments and are happy that this is a much-improved manuscript in its current form. We hope we have answered the reviewers’ and the reviewer editor’s questions and concerns adequately. 

Please feel free to contact me with questions or concerns during your review process.

Sincerely yours,

*Corresponding author: Mubarak Ahmed Mashrah and Liping Wang

Key Laboratory of Oral Medicine, Guangzhou Institute of Oral Disease, Stomatology Hospital of Guangzhou Medical University, Guangdong Guangzhou 510140, China

mubarak198226@gmail.com and wangliplj@126.com Mobile: 0086-13172053275

---

## [Editor Report · Decision Letter 1]

2 Sep 2021

Survival of vascularized osseous flaps in mandibular reconstruction: A network meta-analysis

PONE-D-21-16832R1

Dear Dr. Mashrah,

We’re pleased to inform you that your manuscript has been judged scientifically suitable for publication and will be formally accepted for publication once it meets all outstanding technical requirements.

Kind regards,

Peter Dziegielewski, MD, FRCSC

Academic Editor

PLOS ONE
---

## [Editor Report · Acceptance letter]

13 Oct 2021

PONE-D-21-16832R1 

Survival of vascularized osseous flaps in mandibular reconstruction: A network meta-analysis 

Dear Dr. Mashrah:

I'm pleased to inform you that your manuscript has been deemed suitable for publication in PLOS ONE. Congratulations! Your manuscript is now with our production department. 

Kind regards, 

on behalf of

Dr. Peter Dziegielewski 

Academic Editor

PLOS ONE